# History of Heart Failure in Patients Hospitalized Due to COVID-19: Relevant Factor of In-Hospital Complications and All-Cause Mortality up to Six Months

**DOI:** 10.3390/jcm11010241

**Published:** 2022-01-03

**Authors:** Mateusz Sokolski, Konrad Reszka, Tomasz Suchocki, Barbara Adamik, Adrian Doroszko, Jarosław Drobnik, Joanna Gorka-Dynysiewicz, Maria Jedrzejczyk, Krzysztof Kaliszewski, Katarzyna Kilis-Pstrusinska, Bogusława Konopska, Agnieszka Kopec, Anna Larysz, Weronika Lis, Agnieszka Matera-Witkiewicz, Lilla Pawlik-Sobecka, Marta Rosiek-Biegus, Justyna M. Sokolska, Janusz Sokolowski, Anna Zapolska-Tomasiewicz, Marcin Protasiewicz, Katarzyna Madziarska, Ewa A. Jankowska

**Affiliations:** 1Institute of Heart Diseases, Wroclaw Medical University, Borowska 213, 50-556 Wroclaw, Poland; justyna.sokolska@umw.edu.pl (J.M.S.); anna.zapolska@umw.edu.pl (A.Z.-T.); marcin.protasiewicz@umw.edu.pl (M.P.); ewa.jankowska@umw.edu.pl (E.A.J.); 2Institute of Heart Diseases, University Hospital, 50-556 Wroclaw, Poland; koreszka@usk.wroc.pl (K.R.); anna.larysz@umw.edu.pl (A.L.); wlis@usk.wroc.pl (W.L.); 3Biostatistics Group, Department of Genetics, Wroclaw University of Environmental and Life Sciences, 51-631 Wroclaw, Poland; tomasz.suchocki@gmail.com; 4Department of Anesthesiology and Intensive Therapy, Wroclaw Medical University, 50-367 Wroclaw, Poland; barbara.adamik@umw.edu.pl; 5Clinical Department of Internal and Occupational Diseases, Hypertension and Clinical Oncology, Wroclaw Medical University, 50-556 Wroclaw, Poland; adrian.doroszko@umw.edu.pl; 6Gerontology Unit, Public Health Department, Wroclaw Medical University, 51-618 Wroclaw, Poland; jaroslaw.drobnik@umw.edu.pl; 7Department of Pharmaceutical Biochemistry, Division of Pharmaceutical Biochemistry, Wroclaw Medical University, 50-556 Wroclaw, Poland; joanna.gorka-dynysiewicz@umw.edu.pl; 8Department of Nursing and Obstetrics, Division of Internal Medicine Nursing, Wroclaw Medical University, 51-618 Wroclaw, Poland; maria.jedrzejczyk@umw.edu.pl; 9Clinical Department of General, Minimally Invasive and Endocrine Surgery, Wroclaw Medical University, 50-556 Wroclaw, Poland; krzysztof.kaliszewski@umw.edu.pl; 10Clinical Department of Paediatric Nephrology, Wroclaw Medical University, 50-556 Wroclaw, Poland; katarzyna.kilis-pstrusinska@umw.edu.pl; 11Department of Pharmaceutical Biochemistry, Wroclaw Medical University, 50-556 Wroclaw, Poland; boguslawa.konopska@umed.wroc.pl; 12Clinical Department of Internal Medicine, Pneumology and Allergology, Wroclaw Medical University, 50-369 Wroclaw, Poland; agnieszka.kopec@umw.edu.pl (A.K.); marta.rosiek-biegus@umw.edu.pl (M.R.-B.); 13Clinical Department of Heart Transplantation and Mechanical Circulatory Support, Institute of Heart Disease, Wroclaw Medical University, 50-556 Wroclaw, Poland; 14Wroclaw Medical University Biobank, Wroclaw Medical University, 50-556 Wroclaw, Poland; agnieszka.matera-witkiewicz@umw.edu.pl; 15Division of Basic Sciences, Faculty of Health Sciences, Wroclaw Medical University, 50-368 Wroclaw, Poland; lilla.pawlik-sobecka@umw.edu.pl; 16Clinical Department of Emergency Medicine, Wroclaw Medical University, 50-556 Wroclaw, Poland; janusz.sokolowski@umw.edu.pl; 17Department of Nephrology and Transplantation Medicine, Wroclaw Medical University, 50-556 Wroclaw, Poland; katarzyna.madziarska@umw.edu.pl

**Keywords:** COVID-19, SARS-CoV-2, heart failure, morbidity, mortality, long-term outcome

## Abstract

Background: Patients with heart failure (HF) are at high risk of unfavorable courses of COVID-19. The aim of this study was to evaluate characteristics and outcomes of COVID-19 patients with HF. Methods: Data of patients hospitalized in a tertiary hospital in Poland between March 2020 and May 2021 with laboratory-confirmed COVID-19 were analyzed. The study population was divided into a HF group (patients with a history of HF) and a non-HF group. Results: Out of 2184 patients (65 ± 13 years old, 50% male), 12% had a history of HF. Patients from the HF group were older, more often males, had more comorbidities, more often dyspnea, pulmonary and peripheral congestion, inflammation, and end-organ damage biomarkers. HF patients had longer and more complicated hospital stay, with more frequent acute HF development as compared with non-HF. They had significantly higher mortality assessed in hospital (35% vs. 12%) at three (53% vs. 22%) and six months (72% vs. 47%). Of 76 (4%) patients who developed acute HF, 71% died during hospitalization, 79% at three, and 87% at six months. Conclusions: The history of HF identifies patients with COVID-19 who are at high risk of in-hospital complications and mortality up to six months of follow-up.

## 1. Introduction

Heart failure (HF) is one of the most common causes of death worldwide [1]. Approximately 64.3 million people currently live with HF, and the prevalence is estimated at 1–2% of the adult population in developed countries [2,3]. According to the studies of the HF Working Group of the Polish Cardiac Society based on reports of the National Health Fund, about 1.2 million people suffer from HF and 140 thousand die every year in Poland [4]. Despite the decline in cardiovascular mortality in the last decades, the prognosis in HF remains still poor. Approximately 24% of patients die within one year of HF diagnosis [5]. In-hospital mortality during HF hospitalizations is also unacceptably high and varies from 12 to 19%, depending on consecutive years, gender, and age groups [6].

The outbreak of coronavirus disease 2019 (COVID-19) pandemic caused by severe acute respiratory syndrome coronavirus 2 (SARS-CoV-2) became a global health crisis that resulted in almost 4.9 million deaths worldwide until October 2021, while more than 241 million people were infected, and these numbers are still growing [7]. The clinical course of the SARS-CoV-2 infection range from asymptomatic to critical presentation [8]. While the severe and critical represented 21% of COVID-19 patients, the in-hospital infection fatality rates vary from 11% to 31% [8,9,10]. Three and six months all-cause mortality in patients hospitalized with COVID-19 was estimated at 28% and 30%, respectively [11]. However, the overall mortality rate in the population is lower ranging 0.3–5% [8,12,13].

Numerous risk factors for severe disease and death have been described since the beginning of the pandemic, including age, gender, lifestyle, laboratory indications, complications, and comorbidities [14]. Older age is associated with the risk of unfavorable course of the disease. In the United States, deaths of patients aged 50 years and older account for >94% of the total deaths due to SARS-CoV-2 infection [15,16]. While gender does not affect the risk of infection, in males with COVID-19, poorer outcomes and more deaths are observed, independent of age [17].

Patients with HF are at especially high risk of morbidity and mortality from SARS-CoV-2 infection [18]. COVID-19 is associated with an immune response leading to systemic inflammation [19]. Increased metabolic demands related to severe inflammatory reaction result in cardiac injury and myocardial functional disorders [18]. Moreover, the volume overload secondary to acute kidney injury (observed in up to 30% of patients with COVID-19), symptomatic treatment, fluid administration can aggravate chronic HF as well [13,19,20]. As a result, COVID-19 can exacerbate a pre-existing HF or even cause it de novo [2].

Interestingly, the COVID-19 pandemic has significantly decreased the number of overall hospital admissions for HF [21,22,23]. While patients with HF less frequently self-referred to the hospital, the number of HF patients brought by an ambulance increased [21]. Moreover, during the pandemic, HF patients were less frequently admitted in New York Heart Association (NYHA) Class II compared with the pre-pandemic period [22].

The multicenter, international retrospective cohort study showed that COVID-19 patients with HF are at increased risk for in-hospital death and in-hospital worsening of HF or acute HF de novo, which further increases in-hospital mortality [24].

Despite intensive investigation of cardiovascular diseases in COVID-19, still, little is known about the long-term outcomes of HF patients hospitalized with COVID-19. The aim of this study was to evaluate clinical characteristics, in-hospital course, and posthospital outcomes of this group.

## 2. Materials and Methods

### 2.1. Study Population

The single-center retrospective cohort study included consecutive adult patients hospitalized in the University Hospital, Wroclaw, Poland between March 2020 and May 2021 with laboratory-confirmed COVID-19 defined as a positive result by polymerase chain reaction testing of a nasopharyngeal sample or a positive blood antigen test. The study population was divided into two subgroups:

(a)HF group: Patients with a history of HF, diagnosed before hospitalization, ac-cording to the European Association of Cardiology (ESC) guidelines [25,26],(b)Non-HF group: Patients without previous diagnosis of HF.

### 2.2. End-Points of the Study and Clinical Follow-Up

The primary endpoint was all-cause mortality at three months.

The secondary endpoints were:

(a)in-hospital mortality,(b)duration of hospitalization,(c)admission to intensive care unit (ICU),(d)the incidence of complications during hospitalization: shock, myocardial infarc-tion (diagnosed according to Fourth Universal Definition of Myocardial Infarction [27]), thromboembolic disease, stroke, acute HF,(e)all-cause mortality at six months.

### 2.3. Study Procedures

The demographic data, the information about medical history, previous medication, signs, symptoms at admission, laboratory tests, in-hospital clinical course were extracted from electronic medical records.

The following laboratory measurements were evaluated on admission: hematology: hemoglobin, leukocytes, lymphocytes, neutrophils; serum electrolytes: sodium, potassium; markers of infection: C-reactive protein (CRP), procalcitonin, interleukin-6 (IL-6), ferritin; renal and liver function tests: estimated glomerular filtration rate (eGFR), blood urea nitrogen (urea), uric acid, bilirubin, gamma-glutamyl transferase (GGTP), albumin; cardiac troponin I; plasma N-Terminal Pro-B-Type Natriuretic Peptide (NT-proBNP).

Surviving patients were followed by telephone contact after three and six months. Information was obtained directly from patients, their relatives, or from the hospital system. Government General Registry Office data regarding death were used to complete follow-up.

### 2.4. Statistical Analysis

The categorical variables were expressed as a number of patients in a given categories (with a percentage). The intergroup differences in categorical variables were tested using the χ^2^ test. Continuous variables with a normal distribution were expressed as a mean ($\bar{x}$) (with standard deviation (SD)). Continuous variables with a skewed distribution were described by median (m) (with an interquartile range (IQR)). These variables were ln-transformed in order to normalize their distribution, and ln-transformed values were used for further statistical analyses. The intergroup differences for continuous variables were tested using Student’s *t* test. The associations between the occurrence of acute HF and clinical and laboratory variables were tested using stepwise forward selection based on the Akaike criterion. In the results, we got the model with three independent variables (age, history of HF, and history of myocardial infarction). The associations between the clinical and laboratory variables and survival during 90 days in patients with COVID-19 were established using Cox proportional hazard regression model (both univariable and multivariable models). Variables that were statistically significant in the univariable model were included in the multivariable model. To estimate the effect of history of HF on in-hospital, 3-months, and 6-months mortality, Kaplan–Meier curves were constructed. Differences in survival rates were tested with the log-rank test. The *p*-value < 0.05 was considered statistically significant. Statistical analyses were performed using the R software version 3.5.1 (R Foundation for Statistical Computing, Vienna, Austria) and STATISTICA 13 data analysis software system (StatSoft, Inc., Tulsa, OK, USA).

## 3. Results

### 3.1. Characteristic of Study Population and Difference between Patients with a History of HF and without HF

The study included 2184 patients with a mean age of 65 ± 13 years, 1082 (50%) were male. There were 255 (12%) patients with a history of HF. Patients from the HF group were older, more often males, and had more comorbidities than patients without a history of HF. Before hospitalization, HF patients were more frequently treated with recommended HF medication, as well as calcium channel blockers, alpha-adrenergic blockers, statins, antiplatelet drugs, anticoagulants, antidiabetics, inhaled β2-agonists, anticholinergic drugs, and hemodialysis (see Table 1).

Patients with a history of HF reported dyspnea and chest pain on admission more frequently and their physical examination revealed more abnormalities (lower oxygen saturation on room air, wheezing, pulmonary congestion, and peripheral edema than patients from the non-HF group (see Table 2).

Patients with a history of HF were characterized by higher levels of cardiac (NT-proBNP, troponin I) and inflammatory (leucocytes, procalcitonin, IL-6) biomarkers as well as higher potassium, urea, uric acid, bilirubin levels, and lower number of lymphocytes, hemoglobin concentration, and eGFR (see Table 3).

### 3.2. The Association of HF History with In-Hospital Course and Applied Treatment and Procedures during Hospitalization

There were no statistically significant differences between groups in relation to the number of patients admitted to ICU and ventilated mechanically. However, HF patients were characterized by longer hospitalization and more often worsened during treatment (43% vs. 24%, *p* < 0.001), including the development of shock, myocardial infarction (diagnosed according to Fourth Universal Definition of Myocardial Infarction [27]), and acute HF. These patients more often required passive oxygen therapy, noninvasive ventilation, use of catecholamines, loop diuretics, amiodarone, nitroglycerine, direct oral anticoagulants, acetylsalicylic acid, antibiotics, and interventions like coronary angiography, coronary revascularization, and hemodialysis (see Table 4).

The predictors of acute HF were higher age and history of HF (see Table 5). The multivariable model included variables with a *p*-value < 0.001 in the univariable model.

### 3.3. Outcome

There was high mortality in the studied population with 326/2184 (15%) deaths during hospitalization, 546/2184 (25%) after three months, and 578/1491 (26%) after six months from admission. We found a relationship between history of HF before hospitalization and mortality in COVID-19 patients in relation to all endpoints (see Figure 1 and Figure 2). The HF group had significantly higher mortality assessed in hospital: 89 (35%) vs. 237 (12%), 3-month mortality: 132 (53%) vs. 413 (22%) and 6-month mortality: 141 (72%) vs. 436 (47%), all *p* < 0.001.

After the adjustment of variables that appeared to be significant predictors in univariate Cox analysis, history of HF was revealed to be a predictor of higher risk for three months mortality (see Table 6 and Table 7).

Among patients with a history of HF, significantly higher risk of three-month death was noted in patients ≤ 70 years old and in women ≤ 70 years old when compared with non-HF group. There were no differences in mortality risk in patients with a history of HF divided according to gender (see Figure 3).

Patients who developed acute HF during hospitalization had extremely high mortality: 54/76 (71%) died during hospitalization, 60/76 (79%) at three, and 61/70 (87%) at six months. Analyzing HF-group with available left ventricular ejection fraction (LVEF) measurement (130/255 patients, 51%), the mean LVEF was 46% ± 16%. There were 70 (54%) patients with HF with preserved ejection fraction (HFpEF), 19 (15%) with HF with mildly reduced ejection fraction (HFmrEF) and 41 (32%) with HF with reduced ejection fraction (HFrEF). In a subgroup analysis, patients with HFrEF and HFmrEF had higher in-hospital, 3-, and 6-month mortality, compared to patients with HFpEF (see Table 8).

Similar findings are related to those patients that had to be taken to ICU as compared to those without ICU stay (see Table 9 and Table 10).

## 4. Discussion

Our study for the first time analyzed not only the in-hospital course but also long-term mortality in this group of patients. The presented study, performed among a large cohort of hospitalized patients with COVID-19, showed how history of HF affects the in-hospital course and especially long-term follow-up. One-third of HF patients died during hospitalization and the next one-third died after hospital discharge in the next three months.

Older age, gender, and comorbidities are well-known risk factors of severe course of COVID-19. Patients with HF are usually elderly, with multiple underlying medical conditions increasing their risk of complications and mortality after SARS-CoV-2 infection [28,29,30]. Especially cardiovascular diseases and their risk factors (e.g., hypertension, diabetes mellitus, atrial fibrillation, ischemic heart disease) are common among HF patients [31]. However, HF remains independently associated with in-hospital mortality [24]. In our multivariable analyses, history of HF increased the risk of primary endpoint incidence from 30% to 83%.

The human angiotensin converting enzyme 2 (ACE2) receptor is a gateway for SARS-CoV-2 to enter cells. In the heart, the ACE2 receptors can be found on the surface of cardiomyocytes, cardiac fibroblasts, and endothelia of coronary arteries [32]. Recent studies showed increased levels of ACE2 in patients with HF, especially males consistently with the increased severity of COVID-19 among men [32,33,34].

Myocardial damage in COVID-19 can be caused by endothelial dysfunction and systemic inflammatory response, leading to plaque instability and prothrombotic state (type 1 MI), as well as by mismatch between oxygen supply and/or demand related with respiratory failure (type 2 MI) [35,36,37,38]. However, the role of SARS-CoV-2 in the direct myocardial injury (“myocarditis-like” effect) is still to be determined [36].

As reported in our manuscript, patients with a history of HF were characterized by higher levels of cardiac (NT-proBNP, troponin I) biomarkers and more often developed myocardial infarction. MI was diagnosed according to the Fourth Universal Definition of Myocardial Infarction [27]. However, troponin elevation can also be observed in a high number of clinical conditions, e.g., HF and a septic state [36,39]. It should be emphasized that troponin levels are mostly influenced by baseline characteristics, such as pre-existing HF or coronary artery disease [36,37]. In the study of 355 patients hospitalized with COVID-19, the levels of troponin were significantly higher in the group of patients with chronic coronary syndromes (CCS). Nevertheless, over 89% of significant troponin elevations in our sample population were nonspecific troponin leaks while only 11% had actual non-ST elevation [37].

Therefore, cardiac biomarkers elevation in COVID-19 may be sometimes misleading. Although myocardial injury has been shown to be associated with worse outcomes in COVID-19, a recent study revealed that troponin elevation had a prognostic value in non-CCS patients, while it was not predictive of poor outcomes in patients with CCS [36,39] Further studies are necessary to determine the role of troponin elevation as a prognostic marker.

In our study, significantly more women and men died in the HF group, compared to those in the non-HF group, but it also showed that the ratio of deaths between women and men did not differ in both groups. It should be emphasized that a significantly higher proportion of deaths was in younger people and younger women in the HF group than in those without HF.

The renal impairment secondary to acute kidney injury is observed in up to 30% of patients with COVID-19 [13,20]. While cardiorenal syndrome is a spectrum of well-known disorders in HF, in our study, patients with HF had worse kidney function, but the presence of chronic kidney disease was not associated with the higher risk of acute HF [40]. In the multivariable model, the only risk factors for developing acute HF were more advanced age and a history of previous HF.

In-hospital mortality rate of HF patients hospitalized with COVID-19 presented in our study remains consistent with prior literature. A study based on the multicenter, multinational PCHF-COVICAV registry analyzed 1974 patients hospitalized with COVID-19, HF was present in 256 patients. 1282 (65%) of patients had cardiovascular disease and/or risk factors. In this study, in-hospital mortality in the HF group reached 36% [24].

Furthermore, in the retrospective analysis of 6439 patients hospitalized in New York with COVID-19, the group of patients with HF was smaller (*n* = 422; 7%). The most frequent comorbidities in the whole study group were hypertension (35%), obesity (28%), and diabetes mellitus (23%). Likewise, compared with patients without HF, those with a history of HF were older, had a higher prevalence of comorbidities, and were receiving a greater number of medications. While total mortality was 26%, the risk of mortality among patients with HF was significantly higher than in the group of patients without HF (40 vs. 25%) [41].

In the study of 3080 patients with confirmed COVID-19, a group of 152 (5%) had a previous history of HF. Similar to our study, patients with HF had more cardiovascular risk factors and comorbidities than patients without a history of HF, and the most common were hypertension (86%), dyslipidemia (75%), atrial fibrillation/flutter (52%), and diabetes (40%). The overall mortality rate was 21% and was significantly higher in patients with previous HF (49 vs. 19%) [42].

However, the aforementioned studies analyzed HF patients already hospitalized with COVID-19; therefore, a certain selection bias cannot be excluded. In the study of 31,051 ambulatory patients with COVID-19, 20% of them had pre-existing HF and those had a 30-day mortality and 30-day admission rate of 5% and 19%, respectively. The overall 30-day mortality and 30-day admission rates were 2% and 10%, respectively. Patients with HF had higher comorbidity burden than the group without HF—they were more likely to be smokers (29% vs. 13%), have hypertension (84% vs. 50%), diabetes (54% vs. 28%), myocardial infarction (44% vs. 120%), and atrial fibrillation (29% vs. 4%) [43]. Thus, the profile of comorbidities seems similar in the groups of hospitalized and ambulatory HF patients.

In our study group, the frequency of comorbidities was similar to that reported in the prior literature—the most common concomitant disorders were hypertension (84%), atrial fibrillation (53%), diabetes mellitus (48%), and chronic kidney disease (36%) [41,42]. However, the incidence of atrial fibrillation among hospitalized HF patients is much higher in our study than in ambulatory patients [43]. Contrarily, the prevalence of cardiovascular disease and/or risk factors as well as HF were higher in the PCHF-COVICAV registry [24].

Before the COVID-19 outbreak, the prognosis in HF was unsatisfactory with the one year-mortality of patients after HF diagnosis estimated at 24% [5]. In-hospital mortality varied from 12% to 19%, depending on the study [6]. As our study revealed, SARS-CoV-2 infection significantly worsens in-hospital course and long-term prognosis of HF patients with three-month mortality two times higher than in HF patients without COVID-19. In the study assessing long-term outcomes after HF hospitalization during the COVID-19 pandemic in the group without SARS-CoV-2 infection, the median follow-up was 622 days. The number of patients that died post-discharge for 2020 and 2019 cohorts was 172 (44%) and 321 (34%), respectively [44]. Interestingly, a recent study showed that during the COVID-19 outbreak, there was a significant decrease in acute admissions, including HF, to the cardiology and emergency departments. While the in-hospital mortality in 2020 and 2019 was similar (3.6% vs. 3.9%, respectively), the death rates of patients admitted to emergency departments were four times higher [23].

### Limitations

Our study is limited by its retrospective, single-center character, which may limit its generalizability. Moreover, the study protocol did not include assessment of LVEF, and only small group of patients were evaluated with transthoracic echocardiography (TTE) due to safety reasons and from certain fraction of patients previous LVEF measurement was not available. Therefore, we are not able to stratify all HF population in relation to LVEF. Furthermore, the protocol of our study did not distinguish types of MI.

Thus, further prospective studies are required, not only to confirm the impact of SARS-CoV-2 infections on long-term outcomes of patients with HF but also to assess quality of medical care during COVID-19 pandemic. While the pathophysiology of COVID-19 and the risk of HF are still insufficiently understood, the impact of comorbidities on survival of patients with HF and COVID-19 should be also examined. Further analyses are needed in order to explain higher mortality among patients with HF, regardless of the presence of SARS-CoV-2 infection.

## 5. Conclusions

This retrospective study demonstrates high in-hospital mortality and poor long-term outcomes in patients with a history of HF hospitalized with COVID-19. The development of acute HF during hospitalization is associated with extremely high in-hospital and long-term mortality. Therefore, HF patients should be well and closely controlled; they should be treated both during hospitalization and afterward.

## Figures and Tables

**Figure 1 jcm-11-00241-f001:**
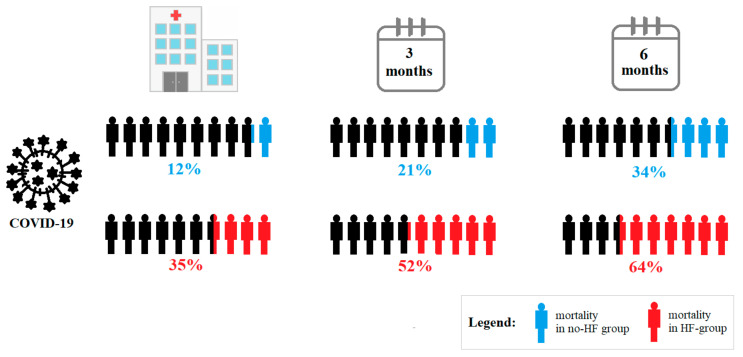
The all-cause mortality in the studied groups.

**Figure 2 jcm-11-00241-f002:**
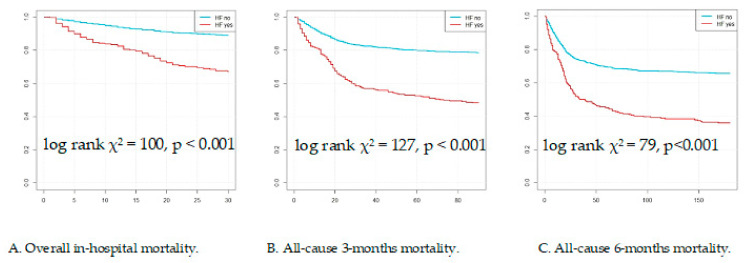
Kaplan–Meier curves for all-cause mortality in studied groups.

**Figure 3 jcm-11-00241-f003:**
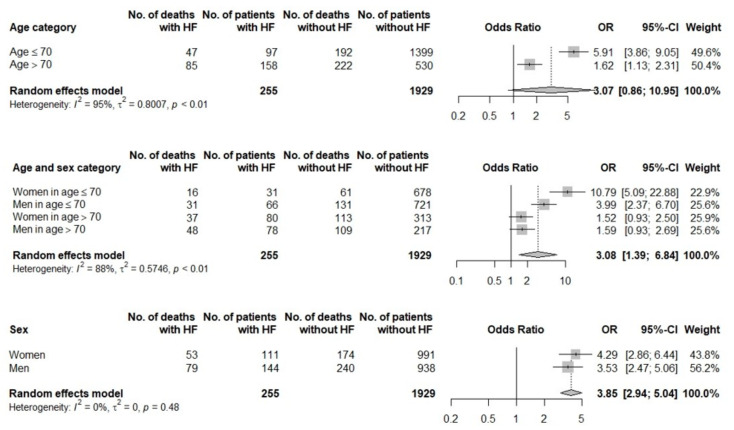
Forest plots for age, sex, and age/sex categories in patients with a history of heart failure vs. without history of heart failure.

**Table 1 jcm-11-00241-t001:** Baseline demographics, clinical characteristics, comorbidities, and related treatment in the studied cohorts.

Variables, Units	HF Group *n* = 255	Non-HF Group *n* = 1929	*p*	Available Data, *n* (%)
**Demographics**				
Age, years	75 ± 12	58 ± 19	<0.001	2184 (100)
Age ≥ 70 years, *n* (%)	169 (66)	571 (30)	<0.001	2184 (100)
Male gender, *n* (%)	144 (57)	938 (49)	0.022	2184 (100)
Body mass index, kg/m^2^	28 ± 6	28 ± 5	0.950	554 (25)
**Co-Morbidities**				
Hypertension, *n* (%)	215 (84)	807 (42)	<0.001	2184 (100)
Diabetes mellitus, *n* (%)	122 (48)	394 (20)	<0.001	2182 (100)
Atrial fibrillation/flutter, *n* (%)	134 (53)	156 (8)	<0.001	2184 (100)
Previous coronary revascularization, *n* (%)	93 (37)	61 (3)	<0.001	2184 (100)
Previous myocardial infarction, *n* (%)	92 (36)	99 (5)	<0.001	2184 (100)
Significant valvular heart disease or previous valve heart surgery, *n* (%)	64 (25)	32 (2)	<0.001	2184 (100)
Peripheral artery disease, *n* (%)	43 (17)	57 (3)	<0.001	2184 (100)
Previous stroke/transient ischemic attack, *n* (%)	53 (21)	111 (6)	<0.001	2184 (100)
Chronic kidney disease or/and hemodialysis, *n* (%)	92 (36)	139 (7)	<0.001	2184 (100)
Asthma, *n* (%)	10 (4)	75 (4)	0.979	2184 (100)
Chronic obstructive pulmonary disease, *n* (%)	29 (11)	46 (2)	<0.001	2184 (100)
Cigarette smoking (previous or current), *n* (%)	55 (22)	138 (7)	<0.001	2180 (100)
Sleep apnea syndrome, *n* (%)	7 (3)	9 (1)	<0.001	2184 (100)
Chronic liver disease, *n* (%)	15 (6)	59 (3)	0.031	2183 (100)
Thyroid disease, *n* (%)	33 (13)/7 (3)	175 (9)/14 (1)	0.001	2184 (100)
Dementia, *n* (%)	50 (20)	82 (4.3)	<0.001	2184 (100)
Malignancy, *n* (%)	29 (11)	156 (8)	0.099	2184 (100)
Transplant recipients, *n* (%)	7 (3)	26 (1)	0.148	2184 (100)
**Treatment Applied before Hospitalization**				
Angiotensin-converting enzyme inhibitors, *n* (%)	104 (41)	248 (13)	<0.001	2184 (100)
Angiotensin receptor blockers, *n* (%)	24 (9)	120 (6)	0.073	2184 (100)
Mineralocorticoid receptor antagonists, *n* (%)	56 (22)	44 (2)	<0.001	2184 (100)
β-blockers, *n* (%)	153 (60)	380 (20)	<0.001	2184 (100)
Ivabradine, *n* (%)	1 (0.4)	2 (0.1)	0.788	2184 (100)
Digitalis glycoside, *n* (%)	10 (4)	9 (1)	<0.001	2184 (100)
Calcium blockers (nondihydropyridine), *n* (%)	11 (4)	27 (1)	0.002	2184 (100)
Calcium blockers (dihydropyridine), *n* (%)	65 (26)	196 (10)	<0.001	2184 (100)
α-adrenergic blockers, *n* (%)	39 (15)	79 (4)	<0.001	2184 (100)
Thiazide or thiazidelike diuretics, *n* (%)	28 (11)	122 (6)	0.009	2184 (100)
Loop diuretics, *n* (%)	100 (39)	85 (4)	<0.001	2184 (100)
Statins, *n* (%)	123 (48)	227 (12)	<0.001	2184 (100)
Acetylsalicylic acid, *n* (%)	76 (30)	182 (9)	<0.001	2184 (100)
Second antiplatelet drug, *n* (%)	20 (8)	19 (1)	<0.001	2184 (100)
Low-molecular-weight heparin, *n* (%)	28 (11)	113 (6)	0.003	2184 (100)
Vitamin K antagonists, *n* (%)	22 (9)	25 (1)	<0.001	2184 (100)
Direct oral anticoagulants, *n* (%)	55 (22)	52 (3)	<0.001	2184 (100)
Insulin, *n* (%)	39 (15)	92 (5)	<0.001	2184 (100)
Metformin, *n* (%)	52 (20)	170 (9)	<0.001	2184 (100)
Sodium–glucose cotransporter-2 inhibitors, *n* (%)	11 (4)	16 (1)	<0.001	2184 (100)
Oral antidiabetics other than mentioned above, *n* (%)	30 (12)	59 (3)	<0.001	2184 (100)
Oral corticosteroid, *n* (%)	9 (4)	83 (4)	0.680	2184 (100)
Immunosuppression other than corticosteroid, *n* (%)	8 (3)	65 (3)	0.993	2184 (100)
Home oxygen therapy or ventilation *n* (%)	4 (1.6)	4 (0.2)	0.005	2184 (100)
Hemodialysis, *n* (%)	21 (8)	37 (2)	<0.001	2184 (100)

**Table 2 jcm-11-00241-t002:** Patient-reported symptoms, vital signs, and abnormalities measured during physical examination at hospital admission in the studied cohorts.

Variables, Units	HF Group *n* = 255	Non-HF Group *n* = 1929	*p*	Available Data, *n* (%)
**Patient-Reported Symptoms**				
Cough, *n* (%)	67 (26)	581 (30)	0.234	2184 (100)
Dyspnea, *n* (%)	144 (57)	777 (40)	<0.001	2184 (100)
Chest pain, *n* (%)	29 (11)	134 (7)	0.016	2184 (100)
Hemoptysis, *n* (%)	4 (2)	11 (1)	0.158	2184 (100)
Smell dysfunction, *n* (%)	4 (2)	72 (4)	0.112	2184 (100)
Taste dysfunction, *n* (%)	5 (2)	61 (3)	0.391	2184 (100)
Abdominal pain, *n* (%)	16 (6)	131 (7)	0.860	2184 (100)
Diarrhea, *n* (%)	16 (6)	111 (6)	0.848	2184 (100)
Vomiting, *n* (%)	11 (4)	87 (4)	0.887	2184 (100)
**Measured Vital Signs**				
Body temperature, °C	36.9 ± 0.9	37.0 ± 0.9	0.182	1186 (54)
Heart rate, beats/minute	85 ± 19	86 ± 16	0.364	1672 (77)
Systolic blood pressure, mmHg	133 ± 26	132 ± 22	0.494	1169 (76)
SpO2 on room air, %	90 ± 10	92 ± 8	0.001	1263 (58)
SpO2 on oxygen supplementation, %	94 ± 6	95 ± 6	0.420	824 (38)
**Abnormalities Detected during Physical Examination**				
Wheezing, *n* (%)	59 (23)	160 (8)	<0.001	2184 (100)
Pulmonary congestion, *n* (%)	85 (33)	282 (15)	<0.001	2184 (100)
Peripheral oedema, *n* (%)	58 (23)	131 (7)	<0.001	2184 (100)

**Table 3 jcm-11-00241-t003:** Laboratory parameters measured on admission in the studied cohorts.

Variables, Units	HF Group *n* = 255	Non-HF Group *n* = 1929	*p*	Available Data,*n* (%)
**Morphology**				
Leukocytes, 10^3^/µL	7.7 (5.8–10.6)	7.3 (5.3–10.3)	0.040	2050 (94)
Lymphocytes, 10^3^/µL	0.9 (0.6–1.4)	1.0 (0.7–1.4)	0.007	1296 (59)
Neutrophils, 10^3^/µL	5.7 (3.8–9.1)	5.5 (3.4–8.2)	0.058	1299 (59)
Hemoglobin, g/dL	12.0 ± 2.4	13.1 ± 2.2	<0.001	2050 (94)
**Electrolytes, Inflammatory**				
Sodium, mmol/L	138 ± 6	138 ± 5	0.958	2032 (93)
Potassium, mmol/L	4.3 ± 0.8	4.1 ± 0.6	<0.001	2039 (93)
CRP, mg/L	46 (13–102)	49 (13–117)	0.686	2020 (92)
Procalcitonin, ng/mL	0.15 (0.06–0.57)	0.08 (0.04–0.26)	<0.001	1475 (68)
IL-6, pg/mL	28 (11–59)	16 (6–44)	0.002	702 (32)
Ferritin, ng/mL	557 (197–1212)	602 (297–1150)	0.309	969 (44)
**Biochemistry**				
Urea, mg/dL	64 (44–101)	36 (25–57)	<0.001	1859 (85)
eGFR, ml/min/1.73 m^2^	53 ± 31	78 ± 34	<0.001	1958 (90)
Albumin, g/L	3.0 ± 0.6	3.1 ± 0.6	0.132	665 (30)
Uric acid, mg/dL	6.7 (5.4–9.2)	5.2 (3.9–6.7)	<0.001	623 (29)
Bilirubin, mg/dL	0.7 (0.5–1.1)	0.6 (0.5–0.8)	<0.001	1408 (64)
GGTP, U/L	43 (24–97)	43 (24–86)	0.916	1352 (62)
**Cardiac Biomarkers**				
NT-proBNP, pg/mL	6496 (2255–15,881)	551 (160–2441)	<0.001	379 (17)
Troponin I, ng/L	55 (23–157)	11 (4–36)	<0.001	1174 (64)

**Table 4 jcm-11-00241-t004:** In-hospital course and therapies applied during the hospitalization in the studied cohorts.

Variables, Units	HF Group *n* = 255	Non-HF Group *n* = 1929	*p*	Available Data, *n* (%)
**In-hospital Course**				
Duration of hospitalization, days	13 (4–21)	9 (2–15)	<0.001	2184 (100)
Admission at intensive care unit, *n* (%)	33 (13)	181 (9)	0.092	2184 (100)
Shock, *n* (%)	38 (15)	150 (8)	<0.001	2184 (100)
Myocardial infarction, *n* (%)	10 (4)	16 (1)	<0.001	2184 (100)
Thromboembolic disease, *n* (%)	10 (4)	59 (3)	0.459	2184 (100)
Stroke, *n* (%)	7 (3)	37 (2)	0.377	2184 (100)
Acute HF, *n* (%)	53 (21)	23 (1)	<0.001	2184 (100)
**Applied Treatment and Procedures**				
Passive oxygen therapy, *n* (%)	109 (43)	654 (34)	0.006	2181 (100)
Non-invasive ventilation, *n* (%)	37 (15)	136 (7)	<0.001	2181 (100)
Mechanical ventilation, *n* (%)	33 (13)	182 (9)	0.098	2184 (100)
Duration of mechanical ventilation, days	8 (2–15)	9 (4–17)	0.176	197 (9)
Therapy with catecholamines, *n* (%)	45 (18)	173 (9)	<0.001	2184 (100)
Therapy with loop diuretics, *n* (%)	104 (41)	229 (12)	<0.001	2184 (100)
Extracorporeal membrane oxygenation, *n* (%)	1 (0.4)	22 (1.1)	0.439	2184 (100)
Coronary angiography, *n* (%)	12 (5)	18 (1)	<0.001	2184 (100)
Coronary revascularization, *n* (%)	11 (4)	20 (1)	0.001	2184 (100)
Hemodialysis, *n* (%)	17 (7)	55 (3)	0.003	2184 (100)
Amiodarone, *n* (%)	17 (7)	39 (2)	<0.001	2184 (100)
Nitroglycerine (i.v.), *n* (%)	7 (3)	12 (1)	0.002	2184 (100)
Low-molecular-weight heparin, *n* (%)	160 (63)	1105 (57)	0.111	2184 (100)
Unfractionated heparin, *n* (%)	12 (5)	110 (6)	0.613	2184 (100)
Direct oral anticoagulants, *n* (%)	29 (11)	6 (3)	<0.001	2184 (100)
Vitamin K antagonists, *n* (%), *n* (%)	5 (2)	15 (1)	0.130	2184 (100)
Acetylsalicylic acid, *n* (%)	81 (32)	285 (15)	<0.001	2184 (100)
Thrombolytic therapy, *n* (%)	1 (0.4)	13 (0.7)	0.911	2184 (100)
Systemic corticosteroid, *n* (%)	129 (51)	967 (50)	0.943	2184 (100)
Convalescent plasma, *n* (%)	33 (13)	206 (11)	0.327	2184 (100)
Hydroxychloroquine/chloroquine, *n* (%)	0 (0)	9 (1)	0.567	2184 (100)
Tocilizumab, *n* (%)	1 (0.4)	24 (1.2)	0.374	2184 (100)
Remdesivir, *n* (%)	38 (15)	305 (16)	0.777	2184 (100)
Antibiotic, *n* (%)	168 (66)	1073 (56)	0.002	2184 (100)
Prone positioning, *n* (%)	22 (9)	182 (9)	0.761	2184 (100)

**Table 5 jcm-11-00241-t005:** Predictors of acute HF during hospitalization in patients with COVID-19.

Variables, Units	Univariable Models	Multivariable Model	
	OR (95% CI)	OR (95% CI)	*p*
Age, per 5 years	1.39 (1.27–1.51)	1.23 (1.11–1.36)	<0.001
Arterial hypertension, yes/no	3.82 (2.24–6.54)	-	-
Diabetes mellitus, yes/no	2.73 (1.72–4.34)	-	-
History of heart failure, yes/no	21.74 (13.05–36.23)	11.42 (6.49–20.10)	<0.001
History of myocardial infarction, yes/no	6.12 (3.71–10.10)	1.67 (0.95–2.94)	0.074
Atrial fibrillation/flutter, yes/no	5.52 (3.45–8.86)	-	-
Valvular heart disease, yes/no	5.58 (3.00–10.38)	-	-
Peripheral artery disease, yes/no	3.84 (1.96–7.53)	-	-
Chronic kidney disease, yes/no	4.53 (2.75–7.46)	-	-

Chi2 = 171, *p* < 0.001. OR: odds ratio, CI: confidence interval.

**Table 6 jcm-11-00241-t006:** Predictors of 3-month mortality—univariable model.

Variables	Units	Univariable Model	
		HR (95% CI)	*p*
Age	5 years	1.31 (1.27–1.35)	<0.001
History of heart failure	yes/no	2.95 (2.42–3.59)	<0.001
Gender	Male	1.53 (1.29–1.82)	<0.001
Diabetes mellitus	yes/no	1.89 (1.57–2.26)	<0.001
Arterial hypertension	yes/no	2.16 (1.82–2.58)	<0.001
Chronic obstructive pulmonary disease	yes/no	2.59 (1.87–3.59)	<0.001
Previous stroke/transient ischemic attack	yes/no	1.80 (1.39–2.33)	<0.001
Chronic kidney disease	yes/no	2.43 (1.97–3.00)	<0.001
Malignancy	yes/no	2.42 (1.93–3.03)	<0.001
C-reactive protein	Ln 1 mg/L	1.49 (1.39–1.60)	<0.001
Hemoglobin	1 g/dL	0.87 (0.84–0.90)	<0.001

**Table 7 jcm-11-00241-t007:** Predictors of 3-month mortality—multivariable model.

Variables	Units	Multivariable Models			
		HR (95% CI)	Wald’s Statistics	*p*	Chi2 (*p*)
Age	5 years	1.29 (1.25–1.33)	269	<0.001	416 (<0.001)
History of heart failure	yes/no	1.54 (1.25–1.88)	17	< 0.001	
Gender	Male	1.53 (1.29–1.82)	20	<0.001	114 (<0.001)
History of heart failure	yes/no	2.86 (2.35–3.48)	109	< 0.001	
Age	5 years	1.32 (1.28–1.37)	283	< 0.001	462 (<0.001)
Gender	Male	1.82 (1.53–2.17)	45	< 0.001	
History of heart failure	yes/no	1.48 (1.21–1.81)	14	< 0.001	
Age	5 years	1.32 (1.27–1.36)	250	< 0.001	504 (<0.001)
Gender	Male	1.71 (1.43–2.04)	36	< 0.001	
Diabetes mellitus	yes/no	1.20 (0.99–1.44)	3.6	0.059	
Arterial hypertension	yes/no	0.91 (0.75–1.11)	0.9	0.346	
COPD	yes/no	1.40 (1.00–1.96)	3.9	0.049	
Previous stroke/TIA	yes/no	0.93 (0.71–1.21)	0.3	0.579	
Chronic kidney disease	yes/no	1.40 (1.11–1.75)	8.1	0.004	
Malignancy	yes/no	1.87 (1.49–2.35)	29	<0.001	
History of heart failure	yes/no	1.31 (1.05–1.63)	5.78	0.016	
Age	5 years	1.30 (1.26–1.35)	204	<0.001	571 (<0.001)
Gender	Male	1.59 (1.33–1.91)	26	<0.001	
Diabetes mellitus	yes/no	1.12 (0.92–1.35)	1.3	0.252	
Arterial hypertension	yes/no	0.93 (0.76–1.12)	0.6	0.433	
COPD	yes/no	1.38 (0.98–1.93)	3.4	0.065	
Previous stroke/TIA	yes/no	0.98 (0.74–1.29)	0.03	0.87	
Chronic kidney disease	yes/no	1.30 (1.03–1.65)	4.7	0.03	
Malignancy	yes/no	1.86 (1.47–2.35)	27	<0.001	
CRP	Ln 1 mg/L	1.42 (1.32–1.52)	93	<0.001	
Hemoglobin	1 g/dL	0.94 (0.91–0.98)	9.2	0.002	
History of heart failure	yes/no	1.35 (1.08–1.69)	7	0.008	

HR: hazard ratio; CI: confidence interval; COPD: chronic obstructive pulmonary disease; TIA: transient ischemic attack; CKD: chronic kidney disease; CRP: C-reactive protein; Ln: natural logarithm.

**Table 8 jcm-11-00241-t008:** Outcomes investigated in patients according to left ventricular ejection fraction.

Variables, Units	HFrEF	HFmrEF	HFpEF	*p*
In-hospital mortality, *n* (%)	18 (44)	8 (42)	14 (20)	0.016
3-month mortality, *n* (%)	26 (63)	13 (68)	19 (27)	<0.001
6-month mortality, *n* (%)	26 (74)	13 (76)	25 (42)	0.002

HFrEF: heart failure with reduced ejection fraction; HFmrEF heart failure with mildly reduced ejection fraction; HFpEF: heart failure with preserved ejection fraction.

**Table 9 jcm-11-00241-t009:** Outcomes investigated in patients admitted to intensive care unit vs. those without intensive care unit stay.

Variables, Units	Intensive Care Unit Stay	Without Intensive Care Unit Stay	*p*
In-hospital mortality, *n* (%)	118 (55)	208 (11)	<0.001
3-month mortality, *n* (%)	132 (62)	414 (21)	<0.001
6-month mortality, *n* (%)	134 (71)	444 (34)	<0.001

**Table 10 jcm-11-00241-t010:** Outcomes investigated in patients admitted to intensive care unit according to history of heart failure.

Variables, Units	HF Group	Non-HF Group	*p*
In-hospital mortality, *n* (%)	25 (76)	93 (51)	0.009
3-month mortality, *n* (%)	27 (82)	105 (58)	0.010
6-month mortality, *n* (%)	27 (87)	17 (68)	0.033

## Data Availability

The data sets used and analyzed during the current study are available from the corresponding author on reasonable request.

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
