# Peer review of "History of Heart Failure in Patients Hospitalized Due to COVID-19: Relevant Factor of In-Hospital Complications and All-Cause Mortality up to Six Months"

_jcm, 2022, doi:10.3390/jcm11010241_

Round 1
Reviewer 1 Report
The authors report their experience in patients hospitalized due to Covid-19. They divided the population according to presence-abscence of heart failure. Heart failure presence resulted in a higher mortality rate, during in hospital stay, at 3 and 6 months after discharege. Development of acute heart failure was a further event assosiated with a poor outcome.
In brief, heart failure history should be seen as a red flag for Covid-19 infection.
The study is well written, the population is relevent and the number of variables reported is significant.
I have only one suggestion:
To add in all tables the number of missing data if any.
In brief, I read this article with interest and I have to congratulate the authors for their relevant work.
Reviewer 2 Report
The authors provide a retrospective analysis of 2.184 patients hospitalised for COVID-19 in a single tertiary centre and analysed outcomes in accordance to the presence or absence of heart failure. Overall, 12% had a history of heart failure and HF was associated with worse short and mid-term outcome irrespective of covariates. The study is interesting but several aspects can be improved to further strengthen the manuscript.
Major comments
- Please add information on patients HF entity (i.e. HFrEF, HFmrEF and HFpEF) and providing stratification of prognosis according to this classification in the supplementary material.
- Please add information on mortality of patients (with and without HF) that had to be taken to ICU as compared to those without ICU stay. The information is clinically relevant and will experience high citations.
- Please add data on vaccination status on patients as well as CT (Crossing threshold) values and provide additional subanalysis on this topic.
Minor comments:
- Some points can be removed from the baseline table (i.e. Peptic ulcer, Proton pump inhibitor)
- ‘To the authors' best knowledge, there are no studies considering long-term outcomes of HF patients hospitalized with COVID-19. Consequently,…’ consider removing this phrase.
- A proper limitations section should be added
- Reference 32 can be removed.
- Language
- ‘higher concentration of infection and organ biomarkers’ likely ‘inflammation and end-organ damage biomarkers’
- ‘In the multivariable model where included variables…’ likely ‘Variables which were statistically significant in the univariable model, were included in the multivariable model’
- ‘The second antiplatelet drug’ likely just ‘second antiplatelet drug’
- ‘HF group had significantly’ likely ‘The Hf group…’
- Some grammar issues in the discussion part of the manuscript please double check.
- Statistics
- Consider showing non-significant p-values with 2 decimals only (instead of 3) as it allows the reader to quicker grasp significant values.
- Table 5 rather provide a stepwise forward model as only ~80 patients had acute heart failure and the model might be overadjusting with the number of variables included.
- Figure 2 provide numbers at risk for the figures. Figure 2a should be retained, Figure 2b+c should be merged and additional a landmark analysis (i.e. exclude all patients that died within the in-hospital) should be provided as third figure.
- ‘In our multivariate analyses’ likely multivariable
Reviewer 3 Report
In the article entitled “History of heart failure in patients hospitalized due to COVID-19: relevant factor of in-hospital complications and all-cause mortality up to six months” Sokolski et al. analyzed data of 2184 patients hospitalized in a tertiary hospital in Poland between March 2020 and May 2021 with laboratory confirmed COVID-19; 12% of them had history of HF. They found that HF patients had longer and more complicated hospital stay, with more frequent acute HF development as compared with non-HF group.
The topic is surely interesting, and the authors should be congratulated for this idea. The strongest point of this work is the inclusion of an out-of-hospital follow-up.
However, some issues have to be addressed:
- In the introduction, it should be worth mentioning that COVID-19 pandemic (especially in the early stage) has significantly decreased overall admissions for HF, as reported in these two studies, that should be briefly mentioned: Severino P, et al. Reduction in heart failure hospitalization rate during coronavirus disease 19 pandemic outbreak. ESC Heart Fail. 2020 Oct 23;7(6):4182–8. doi: 10.1002/ehf2.13043. Epub ahead of print. PMID: 33094929; PMCID: PMC7754919 and Kubica J, et al. Impact of COVID-19 pandemic on acute heart failure admissions and mortality: a multicentre study (COV-HF-SIRIO 6 study). ESC Heart Fail. 2021 Nov 16. doi: 10.1002/ehf2.13680. Epub ahead of print. PMID: 34786869.
- Results: Please rephrase this sentence: “Before hospitalization HF patients were more frequently treated with recommended HF medication, calcium channel blockers, alpha-adrenergic blockers, statins, antiplatelet drugs, anticoagulants, antidiabetics, inhaled β2-agonists, anticholinergic drugs, and hemodialysis (see Table 1).” The authors have indeed mentioned drugs that cannot be considered "HF medications". Beta-blockers, ARBs, ACE-I, sac/val combinations could be mentioned as HF drugs, but not these ones.
- How was myocardial infarction diagnosed during admission? How many patients were invasively studied? How many of them showed a type 1 or 2 MI? I guess that MI diagnosis was mostly based on cTn elevation. However, cardiac troponin elevation is one major point of COVID-19 involvement, especially since myocardial injury is a very controversial topic. Myocardial injury is one of the clinical aspects capable to predict severe COVID-19 (often presenting with decreasing pattern after the acute event) but it could also be elevated in a high number of clinical conditions, including HF and a septic state (as witnessed by the significantly cTn higher levels in the HF groups). Moreover, it should be highlighted that cTn levels are mostly influenced by baseline characteristics, such as pre-existing HF or coronary artery disease, as highlighted in these studies “Redefining the Prognostic Value of High-Sensitivity Troponin in COVID-19 Patients: The Importance of Concomitant Coronary Artery Disease - J. Clin. Med. 2020, 9, 3263; doi:10.3390/jcm9103263” – “Prognostic significance of cardiac injury in COVID-19 patients with and without coronary artery disease. Coron Artery Dis. 2020. doi:10.1097/MCA.0000000000000914.” – “The relationship between coronary artery disease and clinical outcomes in COVID-19: a single-center retrospective analysis. Coron Artery Dis. 2020 Jul 23. doi: 10.1097/MCA.0000000000000934.” Apart from stating how MI was diagnosed, and especially if MI was diagnosis on cTn elevation basis, please discuss these studies and underline how cardiac biomarkers elevation may be sometimes misleading, especially in CCS and/or ischemic chronic HF setting plus an infective state.
- Results: The authors do not report LVEF. This data should be recollected since is crucial to understand clinical outcomes. Please stratify your population accordingly (HFrEF, HFmrEF, HFpEF) and analyze clinical outcomes accordingly as well. I expect that patients with lower LVEF could show worse outcomes. Can you provide this analysis?
- Results: Could the authors recollect NYHA class on admission? This data in HF patients is essential to understand the severity of their baseline condition.
- Results: “After the adjustment of variables that appeared to be significant predictors in univariate Cox analysis, history of HF was revealed to be a predictor of higher risk for three months mortality in multivariable analysis (Table 6)”. I could not see the entire univariate and multivariate analysis to understand which conditions were included in the model and what is their “weight” regarding the primary outcome. It is essential to report these data along with HR and aHR for every single factor included in the model, in order to comment and discuss on all factors that may influence the primary outcome. In the discussion is also reported that: “In the multivariable model, the only risk factors for developing acute HF were more advanced age and a history of previous HF”, which sounds fine, but numbers should be readable. Please entirely report them in a dedicated table.
- Results: The authors state: “Among patients with a history of HF, significantly higher risk of three month death 199 was noted in patients ≤ 70 years old and in women ≤ 70 years old when compared with 200 non-HF group”. Do you think that age may predict better than HF the severity of COVID-19 clinical outcomes, or do you think that there is a strong collinearity between these variables?
- Discussion: The authors state: “Thus, myocardial damage in COVID-19 can be caused by endothelial dysfunction and systemic inflammatory response”. I understand that the authors have tried to briefly summarize this concept and that an extensive discussion may be beyond the scope of this work, but myocardial damage in COVID-19 may happen for several reasons, and also a direct myocardial involvement or a coronary artery involvement due type 1/2 MI - and not only due to a endothelial damage - (as highlighted in this early report on JCM “Acute Coronary Syndromes and Covid-19: Exploring the Uncertainties. J Clin Med. 2020 Jun 2;9(6):1683. doi: 10.3390/jcm9061683. PMID: 32498230; PMCID: PMC7356537.”) may be present. Please briefly expand this point in the discussion, taking a cue from this report.
- Discussion: “It should be emphasized that a significantly higher proportion of deaths was in younger people and in younger women in HF group than in those without HF.” What’s the explanation for this finding? I assume that younger patients suffer from a lower number of comorbidities.
- Discussion: “Yet, the incidence of atrial fibrillation among hospitalized HF patients is much higher in our study than in ambulatory patients”. What is the explanation for this finding? In how many cases AF was a new diagnosis and in how many cases it was pre-existent and COVID-19 elicited the arrhythmia?
- English language is overall fine but some little mistakes/typos/punctuation errors should be fixed e.g. line 228 “confronted with” --> “compared with”. Some sentences would benefit from a rewording to sound more appropriate.
Round 2
Reviewer 2 Report
The authors provide a revised version of their retrospective analysis of 2.184 patients hospitalised for COVID-19 in a single tertiary centre and analysed outcomes in accordance to the presence or absence of heart failure and have added relevant new data.
Major comments
- I acknowledge that there is a certain fraction of patients where LVEF measurement is missing, but authors should try to retain as much LVEF data as possible as they state that HF diagnosis was based on ESC HF guidelines criteria which mainly focus on LVEF at that timepoint! Therefore, they must provide LVEF information (at least at the timepoint of HF diagnosis) for their population (it might also be sufficient to describe HFpEF, HFmrEF, HFrEF, categories rather than absolute LVEF values). NT-proBNP might not be sufficient to discriminate between these cohorts as NT-proBNP levels were relevantly increased in the non-HF group!
